# Role of Spermidine in Photosynthesis and Polyamine Metabolism in Lettuce Seedlings under High-Temperature Stress

**DOI:** 10.3390/plants11101385

**Published:** 2022-05-23

**Authors:** Xin He, Jinghong Hao, Shuangxi Fan, Chaojie Liu, Yingyan Han

**Affiliations:** 1Beijing Key Laboratory for Agricultural Application and New Technique, College of Plant Science and Technology, Beijing University of Agriculture, Beijing 102206, China; hexin2022727@126.com (X.H.); haojinghong2013@126.com (J.H.); 2Beijing Vocational College of Agriculture, Beijing 102442, China; fsx20@163.com

**Keywords:** high-temperature stress, photosynthesis, polyamine metabolism, polyamine inhibitor

## Abstract

High temperature is a huge threat to lettuce production in the world, and spermidine (Spd) has been shown to improve heat tolerance in lettuce, but the action mechanism of Spd and the role of polyamine metabolism are still unclear. The effects of Spd and D-arginine (D-arg) on hydroponic lettuce seedlings under high-temperature stress by foliar spraying of Spd and D-arg were investigated. The results showed that high-temperature stress significantly inhibited the growth of lettuce seedlings, with a 33% decrease in total fresh weight and total dry weight; photosynthesis of lettuce seedlings was inhibited by high-temperature stress, and the inhibition was greater in the D-arg treatment, while the Spd recovery treatment increased net photosynthetic rate (Pn), stomatal conductance (Gs), transpiration rate (Tr), stomatal limit value (Ls), and intercellular CO_2_ concentration (Ci). High-temperature stress significantly reduced the maximum photochemical efficiency (Fv/Fm), photochemical quenching coefficient (qP), electron transport rate (ETR), and photochemical efficiency of PSII (ΦPSII), increased the non-photochemical burst coefficient (NPQ) and reduced the use of light energy, which was alleviated by exogenous Spd. The increase in polyamine content may be due to an increase in polyamine synthase activity and a decrease in polyamine oxidase activity, as evidenced by changes in the expression levels of genes related to polyamine synthesis and metabolism enzymes. This evidence suggested that D-arg suppressed endogenous polyamine levels in lettuce and reduced its tolerance, whereas exogenous Spd promoted the synthesis and accumulation of polyamines in lettuce and increased its photosynthetic and oxidative stress levels, which had an impact on the tolerance of lettuce seedlings.

## 1. Introduction

Lettuce (*Lactuca sativa* L.) belongs to the genus Lactuca in the family Asteraceae and is the most widely distributed leafy vegetable in the world [1,2]. Lettuce is rich in dietary fiber, vitamins, and bioactive compounds [3,4]. The suitable temperature for lettuce growth is 15~20 °C, and high temperatures affect the quality, yield, and marketability of lettuce. Climate change has increased the magnitude and frequency of global temperature extremes (low and high temperatures), leading to disturbances in crop physiological and biochemical changes that affect the entire developmental process of plant growth [5,6]. High-temperature stress is one of the most destructive abiotic stresses that negatively affects crop production [7], and plants exposed to high temperatures for long periods of time eventually die, resulting in economic losses. Therefore, it is particularly important to study the physiological and molecular mechanisms involved in the resistance to high-temperature changes in vegetable crops to improve the tolerance of vegetables [8].

Photosynthesis is a biological process in which plants (including photosynthetic bacteria) convert light energy into chemical energy and perform organic matter synthesis [9]. Photosynthesis is one of the physiological processes in plants most sensitive to temperature changes [10] and temperature is a major factor affecting chlorophyll (Chl) content and plant growth. High-temperature stress negatively affects photosystem I, photosystem II, and dark responses in plants, resulting in reduced photosynthetic electron transport activity and reduced oxygen release [11]. High-temperature stress significantly affects leaf water status, stomatal conductance (Gs), and intercellular CO_2_ concentration (Ci) [12], and the ability to maintain leaf gas exchange and carbon dioxide assimilation rates under heat stress is directly related to thermal tolerance [13].

Polyamines (PAs) are bioactive compounds with low molecular weight and aliphatic nitrogen groups [14] and are ubiquitous water-soluble polycations that play a crucial role in regulating plant physiology and development as well as in increasing plant tolerance to a variety of stress [15,16]. PAs are present in plants in free and conjugated forms [17], but free state PAs are more actively involved in biological processes [18]. The common PAs in plants are putrescine (Put), spermidine (Spd), and spermine (Spm), with Spd having a more pronounced role in plant physiological functions [19,20,21]. In addition, PAs scavenge reactive oxygen species (ROS) and play an important role in regulating plant defense responses to abiotic stresses [15,22]. The center of the polyamine biosynthetic pathway is Put, which is formed by arginine (arg) through three different routes [23,24]. Polyamine inhibitors can interfere with various aspects of polyamine synthesis and thus regulate polyamine levels in plants and have an important role in elucidating the physiological functions of polyamines under adverse stress. Among these studies, the arginine pathway has been demonstrated to be related to the plant response to adversity stress [23]. Therefore, the ADC is considered a stress enzyme. D-arg is a specific inhibitor of ADC [25], which inhibits ADC activity and further affects polyamine synthesis.

At present, exogenous polyamines and polyamine inhibitors have been studied under abiotic stresses, such as low-temperature stress [26,27], salt stress [28], and drought stress [29], and under high-temperature stresses, exogenous polyamines and polyamine inhibitors have also been reported in rice [30], apricot [31], and sweet orange [32]. However, as we know, the effects of exogenous polyamines and the polyamine inhibitor D-arg on lettuce under high-temperature stress have not been reported. Therefore, in this study, the heat-sensitive lettuce variety ‘No. 3 Beisansheng’ was used as the test material, and 1 mM D-arg and 1 mM Spd (the most suitable Spd concentration for lettuce that has been screened in our laboratory) were applied under high-temperature stress [33]. In addition, the physiological mechanism of lettuce seedlings under high-temperature stress was investigated by measuring the changes in endogenous polyamine content in lettuce under high-temperature stress to explain the relationship between polyamine metabolism and high-temperature stress, revealing the internal mechanism of Spd improving the heat resistance of lettuce seedlings, and providing the basis for Spd to alleviate the high-temperature stress of lettuce.

## 2. Results

### 2.1. Effect of Exogenous Spermidine and D-arg on the Growth Indices of Lettuce Seedlings under High-Temperature Stress

As shown in Table 1, the H treatment (35 °C/30 °C, deionized water) increased the height of lettuce compared to the CK treatment(22 °C/17 °C, deionized water), the HD treatment (35 °C/30 °C, 1.0 mM D-arg) decreased the height of lettuce compared to the H treatment, but the difference was not significant, and the HDS treatment (35 °C/30 °C, 1.0 mM D-arg+ 1.0 mM Spd) increased the height of lettuce compared to the HD treatment. The shoot fresh weight and total fresh weight were significantly lower in the HD treatment by 8% and 7%, respectively, compared to the H treatment, while the HDS treatment significantly increased the shoot fresh weight and dry weight as well as total fresh weight and total dry weight compared to the HD treatment. The water contents of H, HD, and HDS treatments were all significantly lower compared to CK, while the differences between these three treatments were not significant. This showed that the growth of lettuce seedlings was significantly inhibited by high-temperature stress, and total fresh weight and total dry weight were significantly decreased, while HD treatment further inhibited the growth of lettuce; however, the HDS treatment alleviated the damage caused by high temperature and D-arg.

### 2.2. Effect of Exogenous Spermidine and D-arg on Malondialdehyde Content and Relative Electrolyte Leakage (REL) of Lettuce under High-Temperature Stress

As seen in Figure 1a, the malondialdehyde (MDA) content of lettuce was significantly higher in the H, HD, and HDS treatments than in CK on the 2nd and 4th days; on day 6 of the treatment, the HDS treatment was significantly lower than the H and HD treatments and not significantly different from CK. This result indicated that the spraying of Spd effectively reduced the MDA content of lettuce. The MDA content of lettuce reached a maximum on the 8th day of the HD treatment, an increase of 14% compared to CK. The maximum MDA content of lettuce was 2.28 μmol·g^−1^ on the 8th day of HD treatment, which was 15% higher than CK. In contrast, the MDA content of lettuce under the HDS treatment was significantly reduced.

Figure 1b shows that the relative electrolyte leakage (REL) of H and HD treatments was higher than the REL of other treatments on the 2nd day, and the HD treatment was significantly higher than the REL of other treatments on the 4th day. However, HDS treatment was significantly lower than HD treatment on the 6th day of treatment, and there was no significant difference from CK. This result indicated that the spraying of Spd effectively reduced the REL. The REL of the HD treatment reached a maximum value of 43 μmol·g−1 on day 8. The maximum REL of the HD treatment reached the maximum value of 43 μmol·g−1 on the 8th day, which was 28% higher than the maximum REL of CK. In contrast, the REL was significantly reduced after Spd spraying.

In conclusion, high-temperature stress led to a significant increase in MDA content of lettuce and REL in lettuce seedlings, which further increased with HD treatment but decreased with HDS treatment. This result indicated that Spd could alleviate the damage caused by high temperature and D-arg.

### 2.3. Effect of Exogenous Spermidine and D-arg on the Chlorophyll Content of Lettuce under High-Temperature Stress

Figure 2 shows the Chl content. Compared with CK, the content of Chla, Chlb, Car, and total Chl under the H treatment showed an increasing trend over time (Figure 2), and the figure shows that the Chla, Chlb, Car and total Chl content of the 3 treatments (H, HD, and HDS) under high-temperature stress from the 2nd day of H treatment was significantly higher than CK. On the 8th day of the high-temperature treatment, Chla, Chlb, Car and total Chl content were significantly lower in the HD treatment than in the H treatment, indicating that D-arg accelerated the degradation of Chl content. However, the Chl content increased significantly after the HDS treatment.

### 2.4. Effects of Exogenous Spermidine and D-arg on the Photosynthetic Parameters of Lettuce under High-Temperature Stress

Net photosynthetic rate (Pn) of lettuce seedlings was significantly reduced under the H treatment compared to CK; Pn was further reduced under the HD treatment compared to the H treatment (Table 2); and Pn recovered under the HDS treatment compared to the HD treatment. Stomatal conductance (Gs) was reduced in the three treatments under the H treatment (H, HD, HDS) compared to the CK treatment, but the differences were not significant. The transpiration rate (Tr) of lettuce seedlings changed insignificantly under the four treatments, with no significant differences. The change in stomatal limitation (Ls) of lettuce seedlings was opposite to the change in intercellular CO_2_ concentration (Ci), with a significant decrease in Ls and a non-significant increase in Ci of lettuce under the H treatment compared to CK; a non-significant difference in Ls and Ci of lettuce under the HD treatment compared to H; and a significant increase in Ls but a non-significant decrease in Ci of lettuce under the HDS treatment compared to HD. The differences were not significant. Water use efficiency (WUE) increased significantly under high-temperature stress compared to CK; no significant difference was found between HD and H treatments, while WUE recovered in HDS-treated lettuce. High-temperature stress inhibited photosynthesis in lettuce seedlings, with greater inhibition in the HD treatment, while the HDS treatment resulted in an increase in Pn, Gs, Tr, Ls, and WUE and a decrease in Ci.

### 2.5. Effect of Exogenous Spermidine and D-arg on Chlorophyll Fluorescence Parameters of Lettuce under High-Temperature Stress

The photochemical quenching coefficient (qP) decreased significantly and the non-photochemical quenching (NPQ) increased significantly under the H treatment compared to CK (Table 3). Under the HD treatment, qP decreased and NPQ increased more significantly compared to the H treatment. The overall trend was reversed with a significant increase in qP and a significant decrease in NPQ with the HDS treatment compared to the HD treatment.

Fv/Fm is the maximum photochemical quantum yield of photosystem II. The Fv/Fm of lettuce under high-temperature stress showed a significant decrease compared with CK, and ΦPSII and ETR also showed the same changes with significant differences. HD treatment decreased Fv/Fm, ΦPSII, and ETR by 6%, 8%, and 37%, respectively, compared to the H treatment, while the decreasing trend was alleviated by HDS treatment.

In conclusion, high-temperature stress significantly decreased Fv/Fm, qP, ETR, and ΦPSII but increased NPQ, which was further suppressed by HD treatment and alleviated by HDS treatment.

### 2.6. Effect of Exogenous Spermidine and D-arg on the Endogenous Polyamine Content of Lettuce under High-Temperature Stress

In Figure 3a, the free Spd content under high-temperature stress was significantly higher than the free Spd content under CK on the 2nd, 4th, and 6th days, and the free Spd content in the HD treatment showed a trend of increasing and then decreasing with the extension of the stress time. The free Spd content of the Spd treatment showed an increasing trend and reached a peak on the 6th day.

The free Spm content under high-temperature stress (Figure 3b) was significantly higher than the free Spm content under CK, and its content gradually increased with treatment time. There was no significant difference in the free Spm content between the H and HD treatments on the 4th, 6th, and 8th days of treatment.

The content of free Put (Figure 3c) was significantly higher under high-temperature stress than under CK on day 8, while HD treatment was decreased by 10% compared to H, but HDS treatment increased it by 6% compared to HD treatment.

The changes in total free polyamine content and free Spd content (Figure 3d) were consistent with each other, and perhaps the Spd content accounted for a larger proportion of the free polyamine content than the Put and Spm content. HD treatment probably inhibited the synthesis of Spd and Put, but the effect on Spm content did not change significantly.

The conjugated Spd content of the HDS treatment was significantly higher than the conjugated Spd content of the other treatments and gradually increased with time (Figure 4a). The conjugated Spm content under high-temperature stress was significantly higher than the conjugated Spm content under CK. On the 4th, 6th, and 8th days, the conjugated Spm content of the H and HDS treatments were significantly higher than the conjugated Spm content of the CK and HD treatments (Figure 4b). The content of conjugated Put was significantly higher under high-temperature stress than under the other treatments, decreased significantly under the HD treatment, and increased again under the HDS treatment (Figure 4c). The total conjugated polyamine content (Figure 4d) was consistent with the change in conjugated Spd content.

The bound Spd content of the H, HD, and HDS treatments (Figure 5a) was significantly higher than the bound Spd content of CK on day 2 of treatment, while the H and HD treatments were significantly higher than the bound Spd content of CK on days 4, 6, and 8 of treatment. There was no significant difference between these 2 treatments, while the HDS treatment increased the bound Spd content, and the difference was significant. The bound-state Spm content (Figure 5b) was higher in the H, HD, and HDS treatments than in CK on days 6 and 8 of the treatment, and the difference in bound-state Spm content was not significant among these three treatments. The bound state Put content (Figure 5c) was significantly lower on day 8 in the HD and HDS treatments than in the CK and H treatments. On days 4, 6, and 8 of treatment, the total bound state polyamine content was significantly higher in the H treatment than in CK (Figure 5d), and there was a small decrease in the HD treatment compared to the H treatment, while the HDS treatment significantly increased the polyamine content.

### 2.7. Effect of Exogenous Spermidine and D-arg on the Polyamine Synthetic and Metabolic Enzyme Activity of Lettuce under High-Temperature Stress

Figure 6 shows the activity of polyamine synthase. The activity of SAMDCase in CK treatment (Figure 6a) remained relatively stable, while H treatment significantly increased the activity of SAMDCase compared to CK. HD treatment decreased the activity of SAMDCase compared to H treatment on days 2, 4, 6, and 8 of treatment, and the difference was significant, while HDS treatment increased the activity of SAMDCase significantly. As shown in Figure 6b, H treatment significantly increased ADCase activity, and HD treatment decreased ADCase activity compared to H treatment on days 4, 6, and 8 of the treatment, and the difference was significant, while HDS treatment increased ADCase activity. As shown in Figure 6c, H treatment significantly increased ODCase activity compared to CK, and HD treatment decreased ODCase activity compared to H treatment on days 4, 6, and 8 of treatment with significant differences, while HDS treatment increased ODCase activity.

The polyamine oxidase activity and polyamine synthase activity in this experiment showed opposite trends (Figure 7). There was no significant difference between the CK and H treatments, while the polyamine oxidase activity of the HD treatment was significantly increased compared to the polyamine oxidase activity of the high-temperature treatment, while the HDS treatment caused a small decrease in polyamine oxidase activity.

### 2.8. Effect of Exogenous Spermidine and D-arg on the Gene Expression of Key Enzymes Involved in Endogenous Polyamine Synthesis and Metabolism in Lettuce

The expression levels of lettuce polyamine synthase-related genes *LsSAMDC*, *LsADC*, *LsSPDS*, and *LsSPS* were significantly upregulated, and the expression levels of polyamine oxidase genes *LsPAO* were significantly downregulated under high-temperature stress (Figure 8), in which the expression of polyamine synthase gene *LsADC* was 7 times higher than the expression of polyamine synthase gene *LsADC* for CK under H treatment and the expression levels were significantly higher than the expression levels of other genes. HD treatment caused a significant increase and decrease in the expression levels of polyamine synthase-related genes and polyamine oxidase-related genes. The expression levels of polyamine synthase-related genes were significantly reduced by HD treatment, and the expression levels of polyamine oxidase-related genes were significantly increased and decreased by spraying Spd treatment.

## 3. Discussion

### 3.1. Effect of Exogenous Spermidine and D-arg on the Growth of Lettuce under High-Temperature Stress

The effects of high-temperature stress on plants vary from species to species [34], with different temperatures, durations, and plant types [35]. Polyamines play a crucial role in regulating plant physiology and development as well as stress [15,16]. The role of Spd in plant physiological functions is more pronounced than the role of other types of polyamines [19,20]. Under drought stress, exogenous spraying of Spd significantly reduced drought stress injury and increased the fresh weight and water content of Fuyun 6. However, exogenous application of polyamine inhibitors further exacerbated the damage caused by drought. This finding is consistent with the phenotypic changes [36]. Under high-temperature stress, the optimal concentration of exogenous Spd is 1.0 mM [33]. Foliar spraying of Spd could reduce the inhibitory effect of high-temperature stress on the aboveground and root growth status of cucumber seedlings and improve biomass accumulation [37]. The results of the present study were in general agreement with the results of previous studies in that the H treatment reduced the total fresh weight, total dry weight, and water content of lettuce seedlings compared to the CK treatment. HD treatment exacerbated the injury caused by high temperature, while HDS treatment increased the biomass accumulation of lettuce seedlings and alleviated the growth inhibition of plants by stress.

### 3.2. Effect of Exogenous Spermidine and D-arg on the Cell Membrane Permeability of Lettuce under High-Temperature Stress

MDA is a product of unsaturated fatty acid peroxidation and is a marker of free radical damage in cell membranes [38]. The maintenance of fatty acid unsaturation plays an important role in the response to environmental stress, and the exogenous application of polyamines reduces lipid peroxidation and ion leakage produced by environmental stress [36,39]. There is evidence that exogenous application of Spd under drought stress attenuated the damage caused by MDA in tea trees, while exogenous polyamine inhibitors significantly increased MDA levels [36]. In addition, overexpression of polyamine oxidase under salt stress was also observed to decrease electrolyte permeability and MDA content, increase peroxidase activity, and alleviate the damage caused by oxidative stress in cucumber transgenic strains [40]. The results of the present study were generally consistent with the results of previous studies, in which H treatment significantly increased the MDA content and electrolyte permeability of lettuce compared with CK, and after HD treatment, these parameters further increased, and HDS treatment caused a significant decrease in MDA and electrolyte permeability, indicating that Spd can alleviate the degree of membrane lipid peroxidation and reduce the damage caused by high-temperature stress on lettuce.

### 3.3. Effect of Exogenous Spermidine and D-arg on Lettuce Photosynthesis under High-Temperature Stress

Photosynthetic pigment content is an indicator of plant stress detection and tolerance [41,42]. High temperature is a major factor affecting chlorophyll content and plant growth. Some studies have shown that the increase in chlorophyll content under high-temperature stress is due mainly to the increase in chlorophyllase activity [9]. In contrast, in the present study, short-term high-temperature stress treatment increased the chlorophyll content of lettuce leaves, which is consistent with previous studies reported in cucumber [43] and may be related to plant species, temperature and polyamine concentration, and treatment time. The slower growth of lettuce seedlings due to short-term heat stress resulted in excessive chlorophyll accumulation per unit leaf area, which caused damage to the leaves [43]. Exogenous spray Spd treatment significantly increased the chlorophyll content, which is consistent with previous findings in citrus, where the exogenous application of Spd to D-arg-treated plants significantly increased the chlorophyll content [28].

Photosynthesis is one of the most temperature-sensitive physiological processes in plants [10], and the mechanisms of response vary among plants under different stresses of the same plant or under the same stress conditions [44,45]. Stomatal and nonstomatal limiting factors play an important role in the reduction of the photosynthetic rate under stress [46]. When intercellular CO_2_ concentration (Ci) and net photosynthetic rate (Pn) change in the same direction and both decrease, the decrease in photosynthetic rate at this time is due mainly to the decrease in stomatal conductance (Gs), which blocks the supply of CO_2_ in the chloroplast and leads to a decrease in photosynthetic rate. Conversely, when net photosynthetic rate (Pn) decreases and intercellular CO_2_ concentration (Ci) increases, the decrease in photosynthetic rate is not due to intercellular CO_2_ concentration (Ci) but to the decrease in intercellular CO_2_ utilization efficiency due to the decrease in photosynthetic activity of chloroplasts, which causes the accumulation of intracellular CO_2_ leading to the increase in intercellular CO_2_ concentration (Ci) [9]. The results of the present experiment were similar to the second case, in that high-temperature stress reduced net photosynthetic rate (Pn), transpiration rate (Tr), stomatal conductance (Gs), and stomatal limit value (Ls) in lettuce seedlings, and these photosynthetic parameters decreased further after HD treatment, while the situation was alleviated after HDS treatment, suggesting that Spd improved the photosynthetic efficiency and thus the heat tolerance of lettuce under high-temperature stress. This result is in agreement with previous results in C_4_ plant awns, indicating that the decrease in photosynthetic rate was due to nonstomatal factors [47]. The increase in photosynthetic rate after HDS treatment in this study may be due to an increase in endogenous polyamine content, which improved its tolerance leading to a recovery in photosynthetic rate, but further confirmation is needed.

The complex relationship between fluorescence kinetics and photosynthesis is key to our understanding of the biophysical processes of photosynthesis [48]. Chlorophyll fluorescence is an important indicator of photosynthetic energy conversion in PSII and is sensitive to environmental stress responses [49]. The maximum photochemical efficiency (Fv/Fm), non-photochemical quenching (NPQ), photochemical quenching coefficient (qP), Photochemical efficiency of PSⅡ (ΦPSII), and electron transport rate (ETR) are the most important chlorophyll fluorescence parameters and are widely used in plant stress physiology studies [50,51]. Fv/Fm is an indicator of the photosynthetic efficiency of PSII [52]. High-temperature damages mainly the PSII of the plant photosynthetic apparatus, and photochemical efficiency directly determines the leaf ΦPSII and is used to detect the effectiveness of chlorophyll for photosynthetic excitation energy [53]. qP reflects the proportion of open PSII reaction centers and represents the energy of photochemical electron transfer [54]. In the present study, the values of Fv/Fm, qP, and ΦPSII were significantly reduced under high-temperature stress, and exogenous spraying of Spd recovered, but the difference was not significant (Table 4), indicating that high temperature led to the inhibition of electron transfer from the primary receptor plastoquinone (QA) to the secondary receptor plastoquinone (QB) on the receptor side of PSII, limiting the oxidation of QA and leading to the destruction of antenna pigments, making the photochemical PSII. These results are consistent with the findings of Shu et al. [48]. The reduced electron transfer efficiency (ETR) leads to the generation of excessive excitation energy, which in turn exacerbates photoinhibition under high-temperature conditions. In contrast, the nonphotochemical burst coefficient (NPQ) is an important physiological process that plants use to dissipate excess absorbed light energy to protect the photosynthetic machinery from damage [55,56]. In this experiment, PSII transfer activity was reduced under high-temperature stress, and the use of light energy was poor. NPQ dissipates excess light energy mainly in the form of heat energy to adapt to the damage caused by high-temperature stress to the plant. These results suggest that high temperature affects the whole process of photosynthesis in plants, which is maintained through increased synthesis and reduced metabolism of endogenous polyamines, thereby increasing their content.

### 3.4. Effect of Exogenous Spermidine and D-arg on the Endogenous Polyamine Content of Lettuce under High-Temperature Stress

Polyamines are involved as secondary metabolites in the local response of plants to abiotic stresses [57], as well as in plant morphogenesis. Currently, exogenous polyamines, inhibitors of polyamine synthesis, and even transgenic approaches have been widely used to study the role of polyamines in plant development and their mechanisms of action. D-arginine (D-arg) is the most important inhibitor of polyamine biosynthesis. In the present study, Spd was exogenously sprayed instead of applying Put because high levels of Spd may be protective against lettuce under high-temperature stress [57], and a recent study reported that the application of polyamine inhibitors in growth medium reduced Spd levels and inhibited shoots and flowering in Arabidopsis [58]. In rice (*Oryza sativa*) [59], wheat (*Triticum aestivum*) [60], creeping hyssop (*Agrostis stolonifera*) [61], and white clover [29] were also found to have increased endogenous PAs due to exogenous application of Spd, thereby enhancing their drought resistance. In the present study, the polyamine content decreased in the order of free > conjugate > bound, and the variation in the three polyamines showed a similar pattern, and the Spd content was the most abundant among the different forms of polyamines, which is consistent with the results of the study of Huang [57]. The endogenous polyamine content of lettuce increased significantly under high-temperature treatment, and the Spd, Spm, and Put content showed an increasing trend with increasing high-temperature time, consistent with the findings in cauliflower [62]. HD treatment led to a decrease in its endogenous polyamine content, which inhibited the growth of lettuce and caused damage, and HDS treatment significantly increased the endogenous polyamine content and alleviated the damage caused by D-arg, which is consistent with previous findings [30] due to the reversal of D-arg-induced damage by exogenous application of Spd rather than by direct counterstress. The increase in bound-state Put content in this assay is similar to the findings of Botella, and the accumulation of intracellular Put may be a result of stress rather than a protective mechanism [63]. Spd treatment reversed the HD treatment, making it statistically consistent with plants treated with H alone.

Furthermore, the causes of changes in endogenous polyamines in lettuce seedlings under high-temperature stress were investigated, and the activities of polyamine synthase and oxidase enzymes were measured. In this experiment, H treatment induced an increase in polyamine synthase activity in lettuce, HD treatment decreased polyamine synthase activity, and HDS treatment further increased polyamine synthase activity, consistent with previous findings in low-temperature-stressed tobacco [30], while high-temperature stress also accelerated the elevation of polyamine oxidase, with HD treatment significantly increasing polyamine oxidase activity and HDS decreasing polyamine oxidase activity, suggesting that the increase in endogenous polyamine content in lettuce under high-temperature stress may be due to the elevation of polyamine synthase activity and the decrease in polyamine oxidase activity. The inhibitory effect of D-arg on ADC activity and growth is similar to the results of the studies on partial reversal of polyamine addition in dwarf pea [64] and apple healing tissue [65]. The results are consistent with the results of the studies on the partial reversal of polyamine addition in dwarf pea [64] and apple healing tissue [65]. D-arg inhibited the activity of lettuce polyamine synthase, while Spd spraying for recovery increased synthase activity and decreased metabolic enzyme activity, thus accelerating the conversion of exogenous to endogenous polyamines and enhancing lettuce resistance.

The changes in endogenous polyamine content were due to changes in the activities of polyamine synthase and polyamine oxidase. The expression of genes related to key enzymes of polyamine anabolism was further determined in this experiment, and it was shown that γ-aminobutyric acid inhibited the activity and gene expression of polyamine oxidase (PAO) and diamine oxidase (DAO) and enhanced the expression of polyamine synthase activity and its related genes, thereby accelerating the accumulation of putrescine, spermidine, and spermine in the outer skin of apple [66]. The accumulation of putrescine, spermidine, and spermine in apple exocarp was accelerated [66]. In this experiment, the expression levels of polyamine synthase-related genes and polyamine oxidase genes in lettuce were significantly upregulated and downregulated under high-temperature stress, in which the expression level of the polyamine synthase gene *L**sADC* was seven times higher than the expression level of the normothermic control and was significantly higher than the expression level of other genes. The expression level of genes related to polyamine oxidase was significantly increased by HD treatment, and the expression level was increased and decreased by HDS treatment. The changes in the gene expression of polyamine anabolic enzymes further verified that the changes in enzyme activity led to the accumulation of polyamine content, which in turn improved the heat tolerance of lettuce, consistent with previous findings in cucumber, where overexpression of polyamine oxidase increased salt tolerance in cucumber seedlings [39].

In summary, it appears that after spraying with D-arg under high-temperature stress, Put content in free and conjugated form and ADC enzyme activity were significantly reduced, and spraying with Spd for recovery treatment as a compensatory mechanism increased Put content to some extent, but there was no significant difference. Spd may not have promoted the increase in Put content; in turn, the metabolic mechanism was switched, and the high level of Spd had a protective effect on the plant and improved its tolerance, but further proof is needed.

## 4. Conclusions

Under high-temperature stress, spraying with D-arg reduced the endogenous polyamine content of lettuce seedlings and reduced their tolerance; whereas exogenous Spd improved the tolerance of lettuce under high-temperature stress by increasing the activity of SAMDC, ODC, and ADC, reducing the activity of PAO and DAO, and promoting the synthesis and accumulation of polyamine content; the increase in endogenous polyamine content further improved the gas exchange and chlorophyll fluorescence parameters of lettuce seedlings, alleviated the damage caused by MDA and REL, and promoted growth (Figure 9).

## 5. Materials and Methods

### 5.1. Plant Materials and Treatments

The heat-sensitive lettuce variety ‘No. 3 Beisansheng’ was selected as the plant material for this study [9], and the seeds were provided by the Leafy Vegetables Innovation Team of the Beijing University of Agriculture. The seeds were soaked in water for 4 h, placed in Petri dishes containing moist filter paper, and then germinated in a light incubator. When the seeds germinated, they were placed on seedling trays containing 1/2 Hoagland nutrient solution for seedling development in a light incubator with a diurnal temperature of 22 °C/17 °C, a relative humidity of 70~75%, and a photoperiod of 12 h/12 h. When 3 leaves appeared, plants with robust and uniform growth were selected for transplanting, transferred into a hydroponic tank with Hoagland nutrient solution, and placed in the Smart Greenhouse of the Beijing University of Agriculture (temperature and humidity conditions were the same as those of the light incubator), and treated when six leaves appeared on the seedlings. Deionized water, 1 mM D-arg and 1 mM Spd (Sigma-Aldrich, Darmstadt, Germany) were sprayed on the leaf surface and leaf back at 8:45 a.m. each day by a small sprayer, and the sprayed liquid adhered to the leaf surface to the extent that it was all wet but did not drip. A total of 4 treatments, 20 plants per treatment, were randomly selected for the experiment in three replicates. The four treatments (the reason we set up these four treatments was that we wanted to inhibit the production of endogenous polyamine content through D-arg and explore the intrinsic mechanisms through exogenous administration of polyamines) were as follows: CK (control): 22 °C/17 °C, deionized water; H: 35 °C/30 °C, deionized water; HD: 35 °C/30 °C, 1.0 mM D-arg; HDS: 35 °C/30 °C, 1.0 mM D-arg+ 1.0 mM Spd.

The experiment was carried out for 8 days, with the main functional leaves collected on days 0, 2, 4, 6, and 8 of the treatment to determine the indicators, and only the photosynthetic indicators were measured on day 8, with each treatment replicated three times under the same experimental conditions.

### 5.2. Biomass Production

Samples were taken on the 8th day of the trial, and 5 lettuce plants were randomly selected. After measuring their plant height, the roots were rinsed with deionized water, the shoots and roots were separated, and then the fresh weight of each part was weighed on an electronic balance. They were then placed in an electric thermostatic drying oven (BGZ-140, Boxun, Shanghai, China) at 105 °C for 30 min, dried at 75 °C to a constant weight, then weighed dry, and the moisture content was calculated.

### 5.3. Determination of Malondialdehyde and Electrolyte Leakage Rates

The thiobarbituric acid method [67] was used to determine the malondialdehyde content. Briefly, 0.5 g of leaf was added to 4 mL of 10% trichloroacetic acid and a small amount of quartz sand and centrifuged at 4000 r/min for 10 min. Two milliliters of the supernatant was then added to 2 mL of 0.6% thiobarbituric acid, mixed well, boiled for 15 min, cooled, and centrifuged at 4000 r/m in. The absorbance values were read at 450 nm, 532 nm, and 600 nm. The malondialdehyde (MDA) content was calculated as follows: μmoL·g^−1^ = [6.45 (A532 − A600) − 0.56A450]V/0.5 g, where V is the volume of the extract.

The conductivity method [68] was used: 0.1 g of fresh lettuce leaves was rinsed with distilled water and placed in a 10 mL test tube, and 10 mL of deionized water was added, and the test tube was left to stand at room temperature for 12 h. The conductivity (R1) was then measured using a conductivity meter (DDs-11, Shanghai, China), followed by boiling in a water bath for 30 min and cooling to room temperature, and the conductivity value (R2) was measured again. The relative conductivity is used to express the permeability of the cytoplasmic membrane. Relative electrolyte leakage (REL) = R1/R2 * 100%

### 5.4. Determination of the Photosynthetic Pigments

The chlorophyll content was determined [69]: 0.3 g of the sample was weighed and put into a mortar and pestle with 2.5 mL of 95% ethanol and ground until homogenized. Then, 10 mL of ethanol was added, and the sample was ground until the tissue turned white and allowed to rest for 3~5 min. The sample was then filtered into a 25 mL brown volumetric flask. The chloroplast pigment extract was poured into a cuvette, and the absorbance of optical density (OD) values of 665, 649, and 470 were measured on a spectrophotometer (UV-5200, Shanghai, China), with 95% ethanol as the blank by taking the extracts and calculating the photosynthetic pigments according to the following equations.
Chl a content (mg·g^−1^) = [13.95OD665 − 6.88OD649]V/1000WChl b content (mg·g^−1^) = [24.96OD649 − 7.32OD665]V/1000WChl content (mg·g^−1^) = [18.08OD649 + 6.63OD665]V/1000 WCarotenoid content (mg·g^−1^) = [1000OD470 − 2.05Chla − 114.8Chlb]/245

### 5.5. Determination of Photosynthetic and Chlorophyll Fluorescence Parameters

Photosynthetic parameters include: net photosynthetic rate (Pn), stomatal conductance (gs), transpiration rate (Tr), intercellular CO_2_ concentration, stomatal limiting values (Ls), and water use efficiency (WUE). The photosynthetic index of lettuce was measured on the 8th day of the treatment from 8:00 am to 11:30 am using a portable photosynthesis meter (CIRAS-3, PP Systems, Amesbury, MA, USA). The room temperature was 25 °C, the light intensity was 600 μmol·m^−2^·s^−1^, the CO_2_ concentration was 400 μmol·mol^−1^ (all CO_2_ was provided by small cylinders) and the relative humidity was 75~80%. Three replicates of each treatment were randomly selected for the experiment. The net photosynthetic rate (Pn), stomatal conductance (Gs), intercellular CO_2_ concentration (Ci), transpiration rate (Tr), limiting value of stomata (Ls), and water use efficiency (WUE) were recorded after the instrumental values had stabilized.

After measuring photosynthetic indicators, chlorophyll fluorescence was measured using a portable photosynthesis meter. For the measurements, each treatment was first dark adapted for 15 min, and the initial fluorescence (Fo) and the maximum photochemical efficiency (Fv/Fm) were first measured. The measurement light was then switched on, and the apparent photosynthetic electron transport rate (ETR), photochemical efficiency of PSII (ΦPSII), photochemical quenching coefficient (qP), and nonphotochemical quenching (NPQ) were measured.

### 5.6. Determination of Endogenous Polyamine Content

The determination of polyamine content in different forms was carried out according to the method of [43,70]. Lettuce leaves (0.5 g, fresh samples) were weighed, placed in a mortar, ground into powder with liquid nitrogen, and then placed in a 10 mL centrifuge tube. Then, 1.5 mL of prechilled 5% perchloric acid was added, vortexed and mixed, extracted in an ice bath for 1 h, and centrifuged at 4 °C and 15,000 rpm for 30 min, after which the supernatant and precipitate were collected. The supernatant was used to determine the free and perchloric acid-soluble bound polyamines; the precipitate was used to determine the perchloric acid-insoluble bound polyamines.

A high-performance liquid chromatography (HPLC) analyzer with a Kro-masd reversed-phase C18 column (250 mm × 4.6 mm) (Agilent, Santa Clara, CA, USA) was used. The mobile phase was set at methanol: water (60:40, *v*/*v*), the flow rate was 0.7 mL·min^−1^, the column temperature was 30 °C, and the detection wavelength was set at 230 nm. The standard curves were plotted with the peak areas as the vertical coordinates and the concentrations of the polyamine standards as the horizontal coordinates, and the correlation coefficients were calculated.

### 5.7. Determination of Key Enzyme Activity for Polyamine Synthesis and Metabolism

Lettuce samples (0.2 g) were placed in a mortar and then ground by adding 2 mL of phosphate-buffered saline (PBS) buffer until homogenized, and the supernatant was taken after centrifugation at 11,000 rpm for 10 min at 4 °C (Centrifuge 5417R, Eppendorf, Germany). S-Adenosylmethionine decarboxylase (SAMDC), arginine decarboxylase (ADC), ornithine decarboxylase (ODC), diamine oxidase (DAO), and polyamine oxidase (PAO) were determined by the method of Gao et al. [24].

### 5.8. Total RNA Extraction and SAMDC, ADC, ODC, PAO, and DAO Transcript Level Analysis

Leaf total RNA extraction was performed using a kit (Yueyang Hua, China). The first strand of cDNA was synthesized using a reverse transcription kit (Tiangen, China). The primer sequences of the relevant genes were downloaded separately from the GenBank library of NCBI. The primer design is shown in Table 4. Quantitative real-time polymerase chain reaction (qRT-PCR) analysis was performed using the product cDNA of reverse transcription as a template, and 18S was used as an internal reference gene. The qRT-PCR was tested using the TB Green Premix Ex Taq II (2×) (Tli RNaseH Plus) Kit (TaKaRa, Japan) on a CFX96 Real-Time PCR Detection System instrument (Bio-Rad Laboratories, Hercules, CA, USA). The reaction system was 10 μL, including 5 μL TB Green Premix Ex Taq II (TliRNaseH Plus), 2 μL ddH2O, 1 μL cDNA template, and 1 μL forward and reverse primers, respectively. The reaction procedure was as follows: 95 °C for 3 min; 95 °C 10 s, 56 °C 30 s, a total of 39 cycles. Gene expression change ploidy analysis was calculated using 2−ΔΔCt, and relative mRNA expression levels were normalized using 18S.

### 5.9. Statistical Analysis

Analysis of trial data was performed using SPSS 24.0 (SPSS Inc. Chicago, IL, USA) and Origin 2017 (OriginLab, Hampton, MA, USA) for analysis of variance and significance. Different letters in the significance analysis indicate significant differences between treatments (*p* < 0.05, Duncan).

## Figures and Tables

**Figure 1 plants-11-01385-f001:**
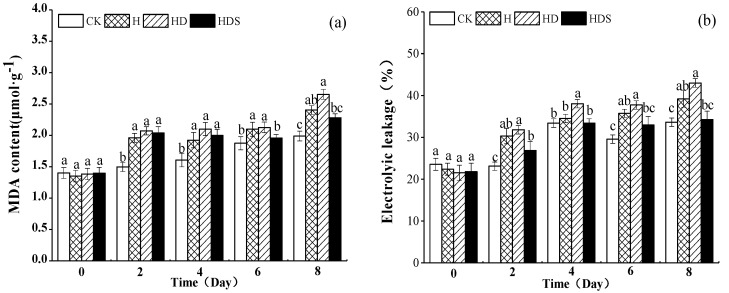
Effects of exogenous spermidine and D-arg on MDA content (**a**) and Relative electrolyte leakage (**b**) of lettuce under high-temperature stress. Vertical bars represent standard deviations of the mean (*n* = 3). Values above each vertical bar followed by different letters show significant differences (*p* < 0.05). CK, 22 °C/17 °C, deionized water; H, 35 °C/30 °C, deionized water; HD, 35 °C/30 °C, 1 mM D-arg; HDS, 1 mM D-arg+ 1 mM Spd.

**Figure 2 plants-11-01385-f002:**
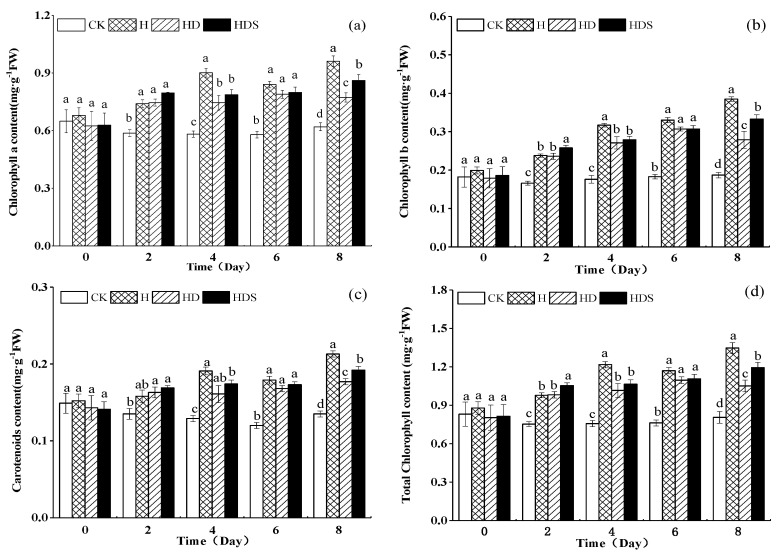
Effects of exogenous spermidine and D-arg on the Chla (**a**), Chlb (**b**), Car (**c**), and total Chl (**d**) content of lettuce under high-temperature stress. Vertical bars represent standard deviations of the mean (*n* = 3). Values above each vertical bar followed by different letters show significant differences (*p* < 0.05). CK, 22 °C/17 °C, deionized water; H, 35 °C/30 °C, deionized water; HD, 35 °C/30 °C, 1 mM D-arg; HDS, 1 mM D-arg+ 1 mM Spd.

**Figure 3 plants-11-01385-f003:**
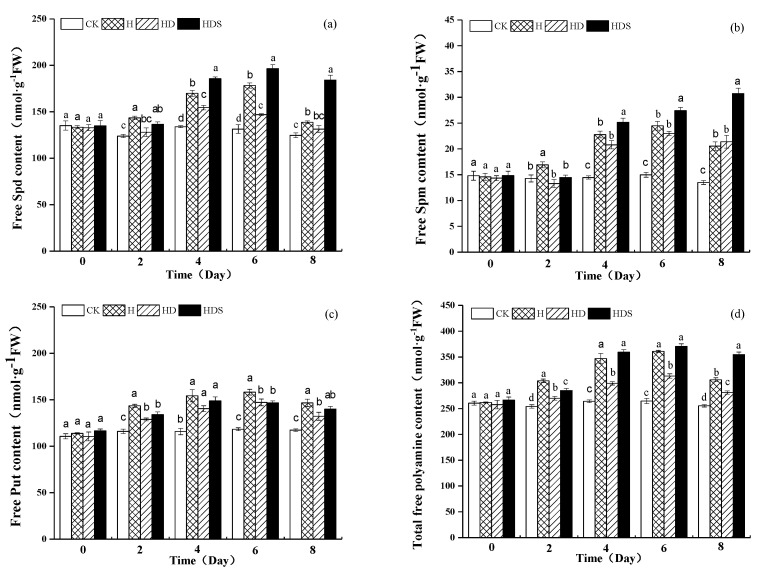
Effects of exogenous spermidine and D-arg on the free Spd (**a**), Spm (**b**), Put (**c**) and Total free polyamines (**d**) contents of lettuce under high-temperature stress. Vertical bars represent standard deviations of the mean (*n* = 3). Values above each vertical bar followed by different letters show significant differences (*p* < 0.05). CK, 22 °C/17 °C, deionized water; H, 35 °C/30 °C, deionized water; HD, 35 °C/30 °C, 1 mM D-arg; HDS, 1 mM D-arg+ 1 mM Spd.

**Figure 4 plants-11-01385-f004:**
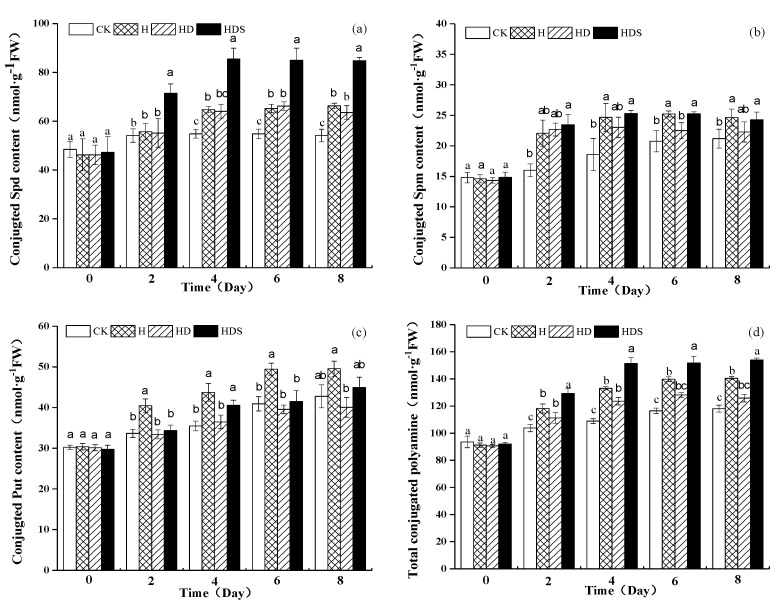
Effects of exogenous spermidine and D-arg on conjugated Spd (**a**), Spm (**b**), Put (**c**) and Total conjugated polyamines (**d**) content of lettuce under high-temperature stress. Vertical bars represent standard deviations of the mean (*n* = 3). Values above each vertical bar followed by different letters show significant differences (*p* < 0.05). CK, 22 °C/17 °C, deionized water; H, 35 °C/30 °C, deionized water; HD, 35 °C/30 °C, 1 mM D-arg; HDS, 1 mM D-arg+1 mM Spd.

**Figure 5 plants-11-01385-f005:**
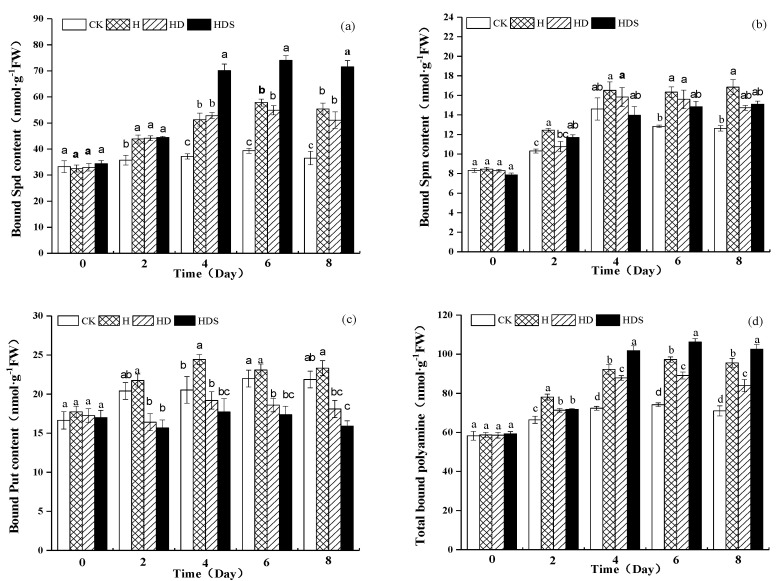
Effects of exogenous spermidine and D-arg on the bound Spd (**a**), Spm (**b**), Put (**c**) and Total bound polyamines (**d**) contents of lettuce under high-temperature stress. Vertical bars represent standard deviations of the mean (n = 3). Values above each vertical bar followed by different letters show significant differences (*p* < 0.05). CK, 22 °C/17 °C, deionized water; H, 35 °C/30 °C, deionized water; HD, 35 °C/30 °C, 1 mM D-arg; HDS, 1 mM D-arg+ 1 mM Spd.

**Figure 6 plants-11-01385-f006:**
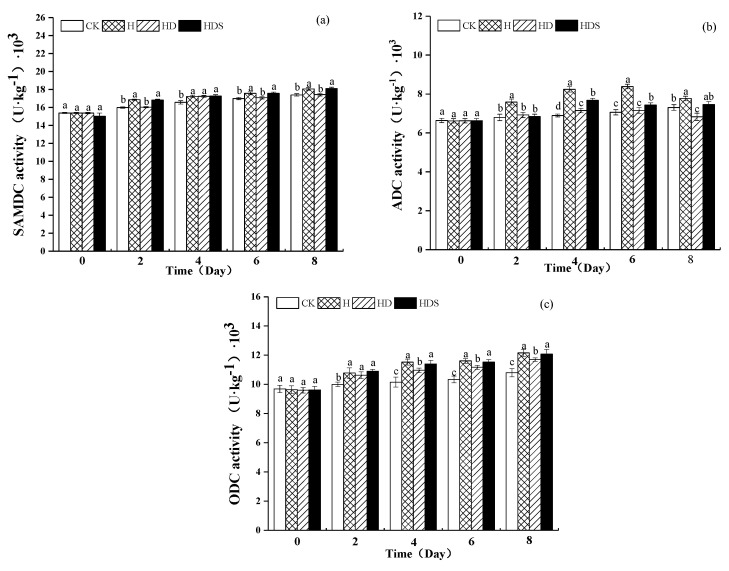
Effects of exogenous spermidine and D-arg on the activities of SAMDC (**a**), ADC (**b**), and ODC (**c**) in lettuce under high-temperature stress. Vertical bars represent standard deviations of the mean (*n* = 3). Values above each vertical bar followed by different letters show significant differences (*p* < 0.05). CK, 22 °C/17 °C, deionized water; H, 35 °C/30 °C, deionized water; HD, 35 °C/30 °C, 1 mM D-arg; HDS, 1 mM D-arg+ 1 mM Spd.

**Figure 7 plants-11-01385-f007:**
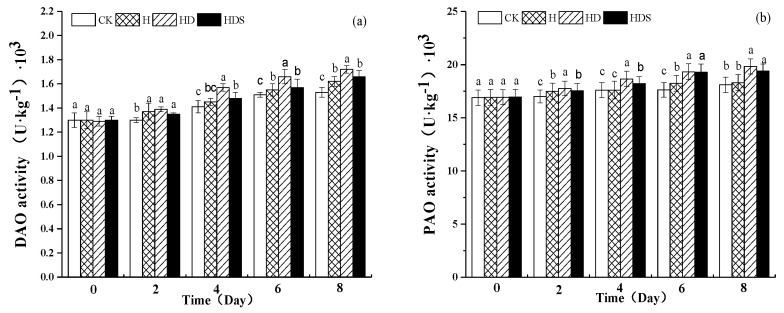
Effects of exogenous spermidine and D-arg on the activities of DAO (**a**) and PAO (**b**) in lettuce under high-temperature stress. Vertical bars represent standard deviations of the mean (*n* = 3). Values above each vertical bar followed by different letters show significant differences (*p* < 0.05). CK, 22 °C/17 °C, deionized water; H, 35 °C/30 °C, deionized water; HD, 35 °C/30 °C, 1 mM D-arg; HDS, 1 mM D-arg+ 1 mM Spd.

**Figure 8 plants-11-01385-f008:**
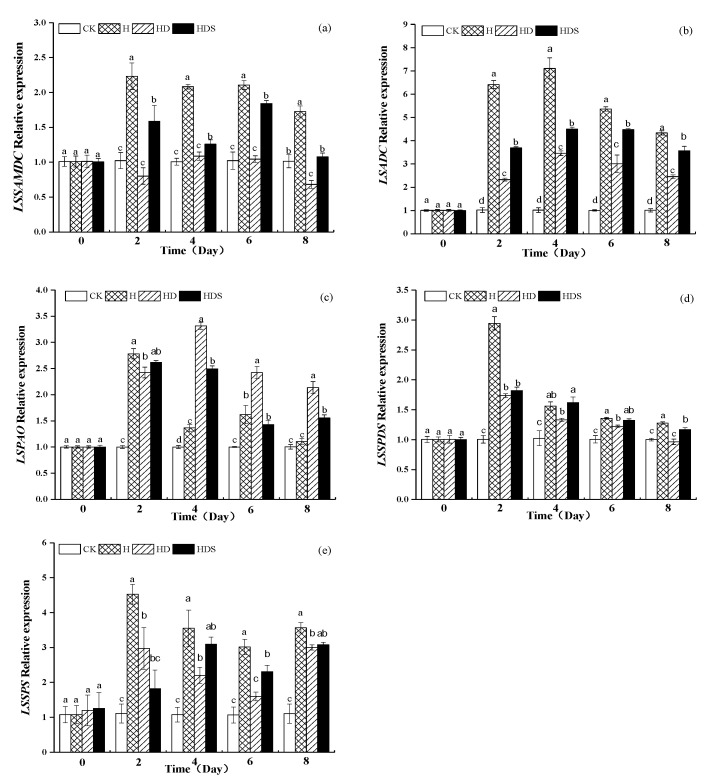
Expression levels of lettuce *LsSAMDC* (**a**), *LsADC* (**b**), *LsPAO* (**c**), *LsSPDS* (**d**), and *LsSPS* (**e**) genes under high-temperature stress. Vertical bars represent standard deviations of the mean (*n* = 3). Values above each vertical bar followed by different letters show significant differences (*p* < 0.05). CK, 22 °C/17 °C, deionized water; H, 35 °C/30 °C, deionized water; HD, 35 °C/30 °C, 1 mM D-arg; HDS, 1 mM D-arg+ 1 mM Spd.

**Figure 9 plants-11-01385-f009:**
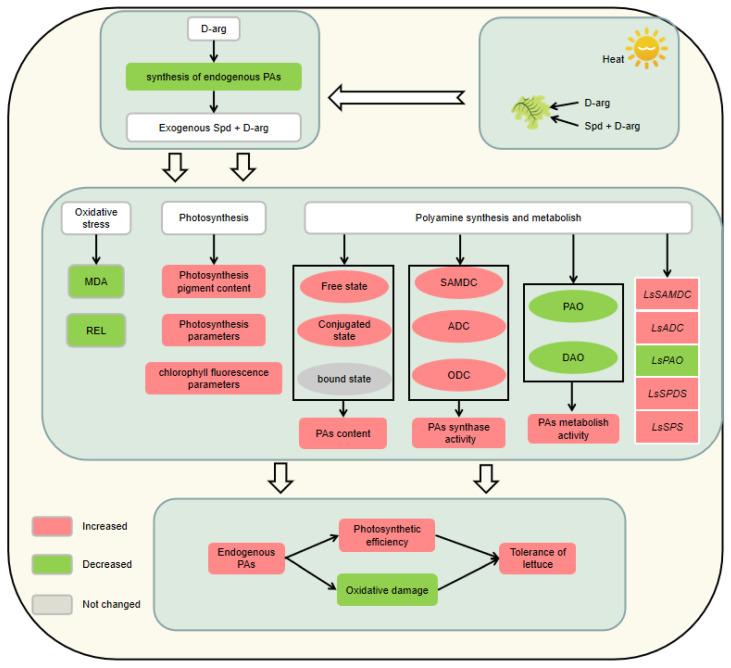
Effect of exogenous D-arg and Spd on lettuce seedlings under high-temperature stress.

**Table 1 plants-11-01385-t001:** Effects of exogenous spermidine and D-arg on growth indicators of lettuce under high-temperature stress.

Treatment	Plant Height (cm)	Shoot		Root		Plant Weight (g)	Dry Weight (g)	Water Content (%)
		Fresh Weight (g)	Dry Weight (g)	Fresh Weight (g)	Dry Weight (g)			
CK	17.93 ± 0.47 b	14.24 ± 1.02 a	0.73 ± 0.06 a	1.78 ± 0.18 a	0.08 ± 0.01 a	16.01 ± 1.19 a	0.81 ± 0.06 a	95.45 ± 0.001 a
H	18.10 ± 0.67 ab	8.55 ± 0.59 b	0.51 ± 0.01 b	0.50 ± 0.02 b	0.02 ± 0.00 c	9.04 ± 0.59 b	0.54 ± 0.01 b	94.25 ± 0.001 b
HD	16.20 ± 0.50 b	7.85 ± 0.18 c	0.52 ± 0.00 b	0.48 ± 0.04 b	0.03 ± 0.00 b	8.33 ± 0.16 c	0.55 ± 0.01 b	93.99 ± 0.001 b
HDS	18.17 ± 0.49 a	10.56 ± 0.49 ab	0.66 ± 0.05 a	0.83 ± 0.09 b	0.04 ± 0.00 ab	11.39 ± 0.54 ab	0.70 ± 0.05 a	94.88 ± 0.002 b

Data are shown as the means, *n* = 3. Values in each column followed by the different letters show significant differences (*p* < 0.05). CK, 22 ℃/17 ℃, deionized water; H, 35 ℃/30 ℃, deionized water; HD, 35 ℃/30 ℃, 1 mM D-arg; HDS, 1 mM D-arg+ 1 mM Spd.

**Table 2 plants-11-01385-t002:** Effects of exogenous spermidine and D-arg on the photosynthetic parameters of lettuce under high-temperature stress.

Treatment	Net Photosynthetic Rate (μmol m^−2^s^−1^)	Stomatal Conductance (mmol m^−2^s^−1^)	Transpiration Rate (mmol m^−2^s^−1^)	Intercellular CO_2_ Concentration (μmol m^−2^s^−1^)	Limiting Value of Stomata	Water Use Efficiency
CK	7.87 ± 0.20 a	117.00 ± 9.61 ab	3.14 ± 0.12 a	313.67 ± 23.62 a	0.30 ± 0.04 a	2.21 ± 0.09 b
H	5.60 ± 0.17 b	105.00 ± 26.51 b	3.07 ± 0.06 a	343.33 ± 10.65 a	0.18 ± 0.02 b	2.53 ± 0.54 a
HD	3.70 ± 0.26 c	90.33 ± 19.92 b	2.22 ± 0.08 a	350.00 ± 10.02 a	0.17 ± 0.03 b	2.57 ± 0.60 a
HDS	4.83 ± 0.19 bc	119.67 ± 27.72 a	3.20 ± 0.07 a	303.33 ± 12.81 a	0.30 ± 0.01 a	2.75 ± 0.57 a

Data are shown as the mean, *n* = 3. Values in each column followed by the different letters show significant differences (*p* < 0.05). CK, 22 °C/17 °C, deionized water; H, 35 °C/30 °C, deionized water; HD, 35 °C/30 °C, 1 mM D-arg; HDS, 1 mM D-arg+ 1 mM Spd.

**Table 3 plants-11-01385-t003:** Effects of exogenous spermidine and D-arg on chlorophyll fluorescence parameters of lettuce under high-temperature stress.

Treatment	qP	NPQ	ETR	ΦPSⅡ	Fv/Fm
CK	0.63 ± 0.03 a	0.31 ± 0.02 d	71.12 ± 1.18 a	0.45 ± 0.09 a	0.79 ± 0.02 a
H	0.34 ± 0.03 b	0.69 ± 0.03 b	64.23 ± 0.91 b	0.18 ± 0.01 b	0.70 ± 0.01 b
HD	0.18 ± 0.01 d	1.11 ± 0.02 a	40.43 ± 0.95 c	0.04 ± 0.01 c	0.66 ± 0.02 c
HDS	0.25 ± 0.01 c	0.41 ± 0.01 c	61.29 ± 0.94 b	0.09 ± 0.00 bc	0.69 ± 0.02 b

Data are shown as the mean, *n* = 3. Values in each column followed by the different letters show significant differences (*p* < 0.05). CK, 22 °C/17 °C, deionized water; H, 35 °C/30 °C, deionized water; HD, 35 °C/30 °C, 1 mM D-arg; HDS, 1 mM D-arg+ 1 mM Spd.

**Table 4 plants-11-01385-t004:** Sequence information of primers of genes of interest used in qRT-PCR assays.

Gene Name	Forwards Primer (5′-3′)	Reverse Primer (5′-3′)	Product Size
18S	GTGAGTGAAGAAGGGCAATG	CACTTTCAACCCGATTCACC	/
SAMDC	TACAATGACGACATGGCGGATAT	CTGGTGGTGGCAACGGAAACTG	80 bp
ADC	AATTAGTCCGCCTTGGTGCTTCC	TACTGCCTGAACAACCGCCATTG	146 bp
SPDS	CCTGATGTAGCGGTTGGATACGAAG	CAGTTCTTGTGCGGGACCTATTGG	144 bp
SPS	ACGGAAAGAGACGAGTTTGCCTATC	CACCATCACCACCACCCACAAC	103 bp
PAO	GCTGCTGGGAGATTCGCTTACG	TATTGAACAGGCTCGGTTGCTTCC	113 bp

## Data Availability

Not applicable.

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
