# Peer review of "Role of Spermidine in Photosynthesis and Polyamine Metabolism in Lettuce Seedlings under High-Temperature Stress"

_plants, 2022, doi:10.3390/plants11101385_

Round 1
Reviewer 1 Report
The subject of this manuscript is interesting, the manuscript is well presented, and the tables and figures are adequate. However, important changes and corrections must be carried out.
Major concerns
There are important grammatical errors and some sentences have to be rewritten. Therefore, English should be carefully revised throughout the manuscript.
The novelty/originality of the manuscriipt is not clear. In the paper by Yang et al. (2022) (ref. 9) authors already concluded that exogenous Spd effectively alleviates damage to lettuce at a high temperature. The data on the effects of Spd on photosynthesis of lettuce seedlings were therefore published. Besides, everry paragraph of the diuscussion ends with "the results are in agreement with previous results ..." Thus, the novelty of the presented data and conclusions are to be specified.
The introduction contains unnessesary well known information with not correct references (for example lines 41-43 with ref. 9). Line 72-73 - 'the effects .... have rarely been reported" - if it was reported even once, please provide a ref.
Lines 43-45 contain very strange statement: photosynthesis is a major factor affecting chlorophyll content.
Line 528 - 5 plants were selecred, but in all tables and figures n=3.
In Results (lines 139-147) the description of the results of chl a and partly total chl content is present, but the chl b and car contents are forgotten.
Fig. 2. Chl content units - mg/kg FW. Obviously the content is calculated per gram of fresh weigh, not per kilogram.
Lines 395-396 and 401-402 are contradictoty.
Minor concerns
Throughout the manuscript latin nameas should be italicized. Some species have latin names, some not.
NpQ - change to NPQ throughout the manuscript
Line 64: ADC - what is it?
Line 87: 'HD treatment increased the height' - in fact the height was decreased
Throughtout the text and especually in subtitles (for exaple on line 104): not MDA, but MDA content. Moreover, MDA content of plants treated by, but not " MDA content of H, HD and HDS treatments".
It is better to use the term RELATIVE ELECTROLYTE LEAKAGE (REL) instead of electrolyte leakage rate, electrolyte permeability, electrolyte permeability rate, electrolyte osmolality.
Table 2. Limiting value of stomata and water use efficiency - not mentioned in the Methods
The significance of differences in some cases is very doubtful - for example in Table 2 for Pn and Gs for HDS treatment and on Fig. 6.
When compare treatments it is not nesessary to provide percents with hundredth (for example 28,84%).
Line 170: Ls - the term was not introduced.
In many cases (for example on lines 178? 193) 'compared to HD treatment" should be added
Line 200: PS II - should be ФPSII
Line 350: bioaccumulation - what does it mean?
Line 413: "fixed chlorophyll" - what is it?
Line 421: Shu et al. - please, provide a reference.
Line 499: "light conversion effeciency" - there was no a word in the text about it.
Line 561: Photosynthetic index - what is it?
Reviewer 2 Report
This presentation needs substantial improvement.
Abstract: So many undefined abbreviations make this uninterpretable on its own. I counted at least 10.
Results: here the text also has many undefined abbreviations - they seem to be defined only in the legends to tables and figures (CK, H, HD, HDS =?)
We need to know the amount of photosynthetically active radiation inside the glasshouse. The photosynthetic rates are so low that it seems like the growth light must have been very dim.
Something is wrong with some of the leaf gas exchange data - the WUE results make no sense compared with photosynthesis and transpiration. In Table 2, the letter indicators of statistical difference make no sense for net Ps.
Reviewer 3 Report
Abstract:
The first sentence of the abstract: ": High-temperature is a huge threat to lettuce production in the world, and the mechanism by which spermidine can improve the heat tolerance of lettuce is unknown." is not an ideal starting point, please re-write it. The role of PAs have been demonstrated in several plant species under different stress condition such as heat, and I think that the authors tried in also in lettuce, but when they started the experiment maybe it was not evident that SPD would be benefitial. (Similar hypothesis needs as in line 68-73)
Also, should be mention why they tested arginine inhibitors (D-arg). For example in order the prove the importance of SPD and the PA metabolism in the stress tolerence?
Please try to avoid the abbreviaitons in the abstract, such as HD treatment or HDS .
What do you mean: "we can conclude that spermidine plays an
important role in polyamine synthesis ..." SPD induced in vivo PA synthesis?
Please rewrite the entire abstract in order to be more focused. The results or conclusion on D-Arg is missing here, one sentence needs.
Introduction:
Lactuca sativa in italics. Asteraceae, too.
Space is missing: " hightemperature"
Please chack the typing error all over the text.
I would not dare to state: "Spd having a more pronounced role in plant 57
physiological functions" based on only 2 citation, however, indeed in some plants, SPD is the most abundant PA.
"The centre of the polyamine biosynthetic pathway is Put, which is formed by arginine (arg)..." you mean from Arg.
"adversity stress [23]. Therefore, the ADC pathway is considered a stress
enzyme associated with adversity. " please try to avoid the repetion : adversity.
Please indicate it D-Arg a reversible or irreversible inhibitor of ADC. It is important information for the choose of the concentration and the treatment frequency what will need.
Is this the real aim of the study: "provide a theoretical basis for exogenous arginine to alleviate high-temperature stress in lettuce."?
Materials and Methods:
Why SPD treatment was not applied alone?
It should be highlighted more that what is the reason why D-Arg +SPD treatment was performed! The application of an ADC inhibitor but the treamtent with SPD, due to the backconversion pathway, could reverse the the lack of PUT synthesis?
"Deionized water, 1 mM D-arg and 1 mM Spd (Sigma-Aldrich, Darmstadt,
Germany) were sprayed on the leaf surface and leaf back at 8:45 a.m. each day.." how many times? only in 5.2 section found the information: "Samples were taken on the 8th day" so i think that for 8 days, according to this 8 times?
"chlorophyll fluorescence" insert "a"
For PA measurement what was the detivarisation agent, the HPLC elution was gradient?
For PA synthesis and catabolism enzym activity measurement the instrument specification is missing.
For the primers please indicate the product size. Also specify what is 18S, as reference gene.
Which PAO gene was investigated, that one which is responsible for the backconverison?
Results:
What can be the reson why the Chl content was higher in heat stressed plants ? The additive negativ effect of heat and D-Arg also missing here?
Discussion is too long, should be shortened in order to highlight the main findings, without repetion of the results.
The conclusion should be re-written to be more focused.
Overall, main concerns:
- the main hypothesis of the study is not enought highlighted, however it could be a novel approach.
- The protective or compensating effect of SPD in combination with D-Arg is not always pronounced.
- The application of D-Arg could not influence the PUT content or ADC activity, it is strange. Although as a compensation mechanism the expression level of ADC indeed increased,and the same is for PAO, suggesting what I mentioned previouly, the lack of PUT induced the metabolic shift in order to the backconversion can compensate it.. (ODC activity slightly increased, too). These finding should be synthesised.
Reviewer 4 Report
Please find attached my comments

Round 2
Reviewer 1 Report
The authors improved the paper and answered questions, but not all. Some corrections are still needed.
- Once again - when compare treatments do not provide percents with hundreds (use 33% instread of 33,33%) - lines 15, 96, 121, 132 et al.
- Line 20 - photochemical
- References 9 and 10 on lines 46, 48 (also 11-13) are not correct ones to use while citating well-known facrs written in textbooks.
- On line 48 chl was introduced, therefore change chlorophyll to chl throughout the text (lines 148-155 et al)
- REL is not introduced in the section 5.3, but is used in results. Don't repeate 'relative electrolyte leakage' on lines 123-136, use REL
- Table 3. The value of Fv/Fm in CK treatment is 0.77 (lower than 0,79), which means that photosynthetic apparatus is not in optimal state and control plants are somehow stessed. How do you explain it?
- Once again - bioaccumulation - not corrected.
- Line 444: PSII transfer activity - what is it?
- Line 445: light energy utilization - how did you calculated it?
- Fig. 9. There are some misprints - parament, paramenters, symthesis
- Lines 517-518 and 525 - the world "with" is missing - spraying with D-arg.
- Line 529 - "increasing their photosynthetic and oxidative stress levels" - it does not correspond to the results and fig. 9
- Line 590: photosynthetic pigments (not chl only) as car was determined in the same samples.
Reviewer 3 Report
The authors made several corrections, and most of their answers can be accepted.
However some other corrections need.
- Again as in the previous round "Put is synthesis by Arg." Please check the synthesis pathwas the name and terms. As ADC is not equal with Arg. Indeed Put is snythesised FROM Arg, in the arginine patway, but by ADC, which the arginine decarboxylase enzyme.
- For the derivatisation you really used methanol? And in which kind of method, as pre-column derivatisation?
- In the abstract, in line 11 please correct: the action mechanism of Spd, and the role of PA metabolism is still unclear.
- Also in the abstract in line 14: Arginine inhibitor (D-arg), do your really mean this? Arginine has no inhibitor, but ADC has.
- Also in the abstract in line 14-16, in the fist part of sentence the "high temperature stress" was repeated twice.
- In line 67 again ADC not equal with Arg!
- 152: "on the 8th of the high temperature treatment" you mean day?
- Has the discussion section bee shortened?
- 517: "spaying D-arg" with is missing
- 518 replase state with form
- 518 spraying Spd, here again with is missing
- 529 "increasing their photosynthetic and oxidative stress levels" do you really mean these?
- gene names in italics.
- the graphycal abstract is not enough representative and understandable, check all the boxes.
Reviewer 4 Report
Please find attached my comments
